# Knowledge Graph Tuning: Real-time Large Language Model Personalization based on Human Feedback

## Abstract

Large language models (LLMs) have demonstrated remarkable proficiency in a range of natural language processing tasks. Once deployed, LLMs encounter users with personalized factual knowledge, and such personalized knowledge is consistently reflected through users' interactions with the LLMs. To enhance user experience, real-time model personalization is essential, allowing LLMs to adapt user-specific knowledge based on user feedback during human-LLM interactions. Existing methods mostly require back-propagation to finetune the model parameters, which incurs high computational and memory costs. In addition, these methods suffer from low interpretability, which will cause unforeseen impacts on model performance during long-term use, where the user's personalized knowledge is accumulated extensively. To address these challenges, we propose Knowledge Graph Tuning (KGT), a novel approach that leverages knowledge graphs (KGs) to personalize LLMs. KGT extracts personalized factual knowledge triples from users' queries and feedback and optimizes KGs without modifying the LLM parameters. Our method improves computational and memory efficiency by avoiding back-propagation and ensures interpretability by making the KG adjustments comprehensible to humans. Experiments with state-of-the-art LLMs, including GPT-2, Llama2, and Llama3, show that KGT significantly improves personalization performance while reducing latency and GPU memory costs. Ultimately, KGT offers a promising solution of effective, efficient, and interpretable real-time LLM personalization during user interactions with the LLMs.

## 1 Introduction

Large language models (LLM) have shown increasing power on various NLP tasks (Devlin et al., 2018; Raffel et al., 2020; Brown et al., 2020; Fedus et al., 2022; Zhang et al., 2021; Zeng et al., 2021). The development of LLMs typically encompasses several key phases (Hyperight, 2024): pre-training on vast corpora, alignment to ensure adherence to ethical guidelines, and domain-specific fine-tuning. While these steps are crucial, there is often an oversight in recognizing the need for further personalization during the deployment phase. Once deployed, LLMs encounter users with personalized factual knowledge (Petroni et al., 2019; Jiang et al., 2020; Roberts et al., 2020). These individual knowledge bases are consistently reflected through users' feedback on various queries and the model's responses during interactions, as shown in the interaction example in Figure 1. To enhance user experience, real-time model personalization is essential, allowing LLMs to adapt and incorporate user-specific knowledge based on persistent feedback throughout human-LLM interactions.

Several technologies are applicable to personalize the LLM, including Parameter Efficient Finetuning (PEFT) (Ding et al., 2023), Knowledge Editing (KE) (Geva et al., 2020; Dai et al., 2022; Meng et al., 2022), and in-context learning (Brown et al., 2020). Nevertheless, these methodologies are often plagued by low efficiency, poor interpretability, or a combination of both drawbacks. PEFT and KE require back-propagation to optimize the model parameters. Back-propagation incurs unacceptable GPU memory and computational costs for the daily use of LLMs, especially for on-device applications where the onboard resources are limited. The high computational overhead also makes it difficult to realize real-time personalization. In addition to the low efficiency, these parameter-based methods also lack interpretability. Modifying the model parameters to satisfy the user's personalized need for

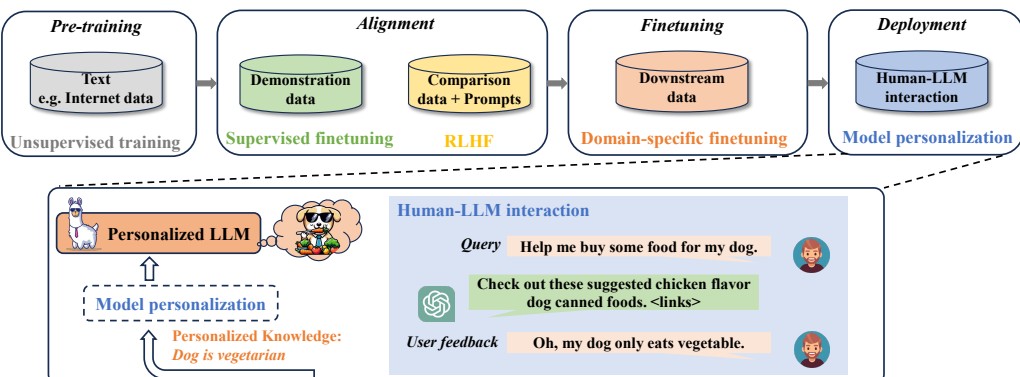

Figure 1: Pipeline of the development of an LLM. In the deployment phase, the model is personalized based on human feedback during the human-LLM interactions. **Note that our method focuses on the penalization that is realized by editing the specific factual knowledge rather than aligning the LLM to some general preference, which should be done in the *alignment* step.** The LLM in the figure is personalized to adapt to the knowledge that the user's dog is vegetarian from the interaction. Then, in the later interactions, the LLM agent will recommend vegetarian dog food for the user given the same query.

the current query may lead to corruption in model parameters, adversely affecting the responses to other queries. This adjustment fails to meet the long-term needs of users who rely on the accumulation of extensive personalized knowledge during interactions with large language models (LLMs). In-context learning has higher interpretability and does not need back-propagation, but its computational overhead, memory cost, and response latency increase drastically with the length of the reference context. Therefore, it is neither efficient nor scalable for long-term use by users.

Knowledge graphs (KGs) (Speer et al., 2017; Sap et al., 2019), which store factual knowledge in a structured format, are increasingly integrated into LLMs to enhance the inference with external knowledge (Chang et al., 2021; Xu et al., 2021; Yao et al., 2022; Song et al., 2021) and are becoming a standard component of LLM systems. In this paper, we propose a new paradigm of model personalization by tuning the KGs based on user feedback. Our proposed method, Knowledge Graph Tuning (KGT), extracts the personalized knowledge triples from the query and the user's feedback. Instead of optimizing the LLM parameters, KGT optimizes the KGs based on the extracted personalized factual knowledge. We formulate the training objective based on the *evidence lower bound (ELBO)* and derive an optimization goal of achieving a *high personalized knowledge retrieval probability* and a *high knowledge-enhanced reasoning probability*. We propose a heuristic optimization algorithm to finetune the KG by adding and removing knowledge triples without touching the model parameters, significantly improving computational and memory efficiency. Additionally, the added and removed knowledge triples are comprehensible to humans, ensuring the method's interpretability.

We conduct experiments on multiple datasets using SOTA pre-trained LLMs. The results demonstrate that KGT significantly improves personalization performance compared with baselines while reducing the latency and GPU memory cost by up to $84\%$ and $77\%$, respectively. By proposing KGT, we offer an effective, efficient, and interpretable solution for realizing real-time LLM personalization during human-LLM interactions. We summarize our contributions as follows:

- We present the necessity of model personalization during the human-LLM interaction and the challenges of adopting existing technologies: low efficiency and poor interpretability.

- We propose a method, KGT, that realizes real-time LLM personalization based on user feedback by optimizing the knowledge graph rather than the model parameters.

- We conduct experiments on multiple datasets with SOTA pre-trained LLMs. Compared with baselines, KGT achieves significantly better personalization performance while reducing computational and memory costs. KGT demonstrates considerable scalability as the volume of queries increases, a critical attribute for fulfilling the long-term needs of users who require accumulating extensive personalized knowledge during their interactions with LLMs.

## 2 RELATED WORKS

**KG-enhanced LLM** Large language models (LLMs) are acclaimed for their proficiency in assimilating knowledge from expansive corpora and achieving groundbreaking results in numerous natural language processing (NLP) tasks. Nonetheless, these models are frequently critiqued for the hallucination issues (Lee et al., 2018; Bang et al., 2023) and their lack of interpretability. In response to these shortcomings, the integration of knowledge graphs (KGs) with LLMs has been proposed (Chang et al., 2021; Xu et al., 2021; Yao et al., 2022; Song et al., 2021). KGs, by storing extensive knowledge in a structured and explicit format, enhance the knowledge capacity of LLMs. Some approaches (Zhang et al., 2019; Rosset et al., 2020) advocate for incorporating KGs during the **pre-training phase** of LLMs, enabling direct knowledge acquisition from these graphs. Others (Chang et al., 2021; Xu et al., 2021) suggest the use of KGs during the **inference phase** to improve access to domain-specific knowledge, thereby substantially enhancing the models' performance. Furthermore, to advance the interpretability of LLMs, researchers are utilizing KGs to elucidate both the factual knowledge (Petroni et al., 2019) and the reasoning processes (Lin et al., 2019) of these models. In this paper, we focus on the setting in which KG enhances the inference phase of an LLM.

**LLM Personalization and Knowledge Editing** Recently, Knowledge Editing (KE) approaches have been proposed to personalize LLMs by localizing and modifying the factual knowledge within transformers. Geva et al. (2020) suggests that the MLP layers within a transformer for masked language models serve as memory units for entities and their associated information. Extending this idea, KN (Dai et al., 2022) developed a technique for updating facts in BERT by manipulating specific rows of the MLP matrix with the embedding of the target object. They pinpoint crucial neurons for storing knowledge through gradient-based methods. De Cao et al. (2021) implement a hyper-network that predicts necessary weight adjustments during test time to modify facts. Their experiments involve BERT and BART (Lewis et al., 2019), particularly in models tailored for question-answering tasks. Mitchell et al. (2022) introduce a hyper-network approach that adjusts gradient decomposition terms to efficiently update knowledge, showcasing scalability across large models like T5 (Raffel et al., 2020) and GPT-J (Wang & Komatsuzaki, 2022). The ROME method (Meng et al., 2022) employed causal tracing to pinpoint knowledge-relevant layers and then edit its FFN module, achieving superior outcomes. All these techniques employ gradient-based optimization methods, which are inefficient and unsuitable for achieving real-time personalization under constrained computational resources. Moreover, modifying model parameters lacks interpretability and may lead to unforeseen impacts on model performance. In contrast, our method solely necessitates model inference, which is significantly more efficient. By adjusting the knowledge graph (KG), our approach ensures that the personalization process is interpretable and the modifications in model performance are comprehensible. There are some recent works (Baek et al., 2024; Madaan et al., 2022) realizing personalization through building a personalized knowledge base, which is similar to our work. However, these works employing unstructured knowledge bases have trouble resolving conflicts in knowledge bases, while our work can detect conflicting knowledge facts during the LLM-AI interaction and update the stale knowledge in the knowledge base.

## 3 PRELIMINARY

**Knowledge Graph (KG)** A knowledge graph $\mathcal{G}$ is a set of triples containing factual knowledge: $\mathcal{G} = \{(e, r, e')|e, e' \in \mathcal{E}, r \in \mathcal{R}\}$, where $\mathcal{E}$ and $\mathcal{R}$ denote the set of entities and relations, respectively, and the triple $(e, r, e')$ is referred to as a knowledge triple.

**KG-enhanced LLM** A KG-enhanced LLM $f_{\theta,\mathcal{G}}$ is parameterized by both the LLM parameters $\theta$ and a KG $\mathcal{G}$. Given a natural language query $q$, the KG-enhanced LLM generates an answer $f_{\theta,\mathcal{G}}(q)$ based on both the model parameters and the retrieved knowledge triples from $\mathcal{G}$. We follow previous works (Sun et al., 2019; Jiang et al., 2022) and assume that the entities $e_q$ in the query and $e_a$ in the answer are labeled. In this paper, we focus on the setting where the LLM retrieves one-depth triples.

## 4 METHOD

This section introduces our method, KGT (Knowledge Graph Tuning), which enables real-time LLM personalization based on user feedback. Instead of finetuning the model parameters, we edit the user's **personalized knowledge graph**, which provides the user's personalized factual knowledge to enhance the customization capability of the LLM. KGT does not need to conduct back-propagation of the LLM, and only the inference is required, which significantly reduces the latency and computational cost. The edited knowledge triples are comprehensible, ensuring the method's interpretability. The overview of KGT is shown in Figure 2.

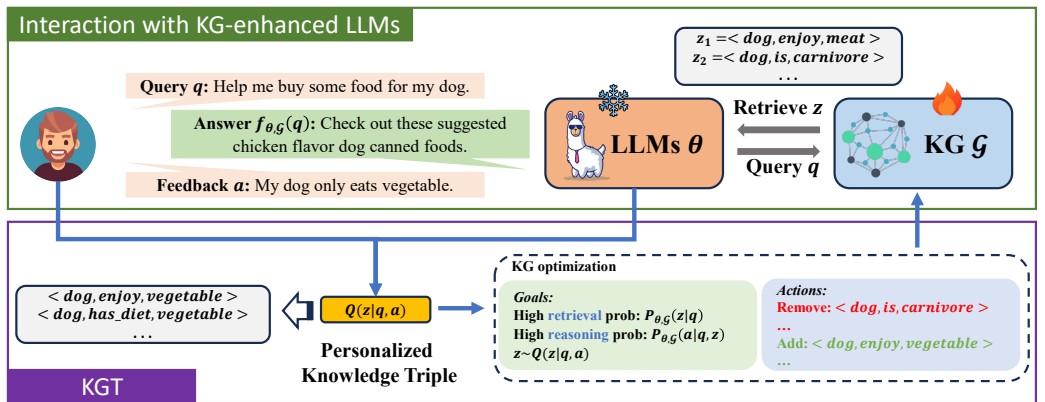

Figure 2: The overview of KGT. The LLM extracts the posterior distribution of the personalized knowledge triples $Q(z|q, a)$ from the human-LLM interaction. The personalized triples are utilized to optimize the KG to achieve two goals: The model can (1) retrieve the personalized triples with high probability and (2) generate the user's feedback with the retrieved triples in high confidence.

### 4.1 KNOWLEDGE GRAPH TUNING

Recently, many techniques have been developed to efficiently finetune or personalize the pre-trained LLM parameters based on the user's data (Ding et al., 2023; Geva et al., 2020; Brown et al., 2020). Instead of finetuning the model parameters, we modify the user's KG to inject personalized knowledge into the LLM.

> **Example** A user wants to let the LLM-based assistant order some food for her vegetarian dog. Instead of finetuning the LLM to remember the fact that the user's dog is vegetarian, the user can simply delete the knowledge triple $(Dog, Enjoy, Meat)$ and add a triple $(Dog, Enjoy, Vegetable)$ in his/her personal knowledge graph. To generate an appropriate response, the LLM assistant will first retrieve the knowledge triple $(Dog, Enjoy, Vegetable)$ based on the user's query, then recommend vegetarian dog food for the user based on the query and the retrieved personalized knowledge triple.

From the example and previous works (Xu et al., 2021; Chang et al., 2021; Dai et al., 2022; Luo et al., 2024), we conclude that there are two steps in KG-enhanced LLM reasoning: 1. knowledge retrieval and 2. knowledge-enhanced reasoning. Following this insight, we formulate the KG-enhanced LLM reasoning probability by marginalizing the knowledge triple distribution. Then, our KGT is an optimization problem that aims to maximize the probability of reasoning the answer from a KG-enhanced LLM:

$$P_{\theta, \mathcal{G}}(a|q) = \sum_{z \in \mathcal{G}} P_{\theta, \mathcal{G}}(a|q, z) P_{\theta, \mathcal{G}}(z|q), \qquad (1)$$

where $P_{\theta, \mathcal{G}}(z|q)$ is the probability of retrieving the knowledge triple $z$ given the user's query $q$, which represents the **knowledge retrieval** step. Let $a$ denote the user feedback, and $P_{\theta, \mathcal{G}}(a|q, z)$ is the probability of generating the answer $a$ given the user's query and the retrieved triple $z$, which stands for the **knowledge-enhanced reasoning** step. Instead of optimizing the model parameters $\theta$, we finetune the knowledge graph $\mathcal{G}$ in KGT.

## 4.2 OBJECTIVE FORMULATION

The training objective in Equation (1) can be optimized by maximizing the evidence lower bound (ELBO) (Jordan et al., 1999), which is formulated as

$$\log P_{\theta,\mathcal{G}}(a|q) \geq \underbrace{\mathbb{E}_{z \sim Q(z)}\left[\log P_{\theta,\mathcal{G}}(a|q,z)\right]}_{\text{knowledge-enhanced reasoning}} \underbrace{-D_{KL}\left(Q(z)||P_{\theta,\mathcal{G}}\left(z|q\right)\right)}_{\text{knowledge retrieval}}, \tag{2}$$

where $Q(z) \triangleq Q(z|q,a)$ is the posterior distribution of $z$ given the user's query and the user's feedback $a$. This posterior distribution gives a larger probability to the triples containing personalized factual knowledge based on the user's feedback (e.g., the example $(Dog, Enjoy, Vegetable)$ in Section 4.1), which are referred to the **personalized triples**. The later term minimizes the KL divergence between the posterior distribution and the prior distribution of the knowledge triple retrieved by the LLM. The former term maximizes the expectation that LLM generates correct answers based on the knowledge triples sampled from the posterior distribution. We notice that there is a related work RoG (Luo et al., 2024) deriving a similar objective with our paper, which is normal since we both derive the objective following the common practice in retrieval-based works (Paranjape et al., 2021; Lewis et al., 2020). However, the subjects of optimization are different in the two papers. RoG trains the model parameter $\theta$ to enhance the LLM's capacity of reasoning on the graph, while we optimize the KG $\mathcal{G}$ to realize efficient and interpretable personalization.

**Knowledge Retrieval**    The goal of the knowledge retrieval term is to finetune the KG such that the LLM can retrieve personalized triples based on the user's feedback. Given a query $q$ and the user's feedback $a$, we can ask the user or utilize the LLM $f_{\theta,\mathcal{G}}$ itself to extract the top-$K$ appropriate relations $\{r_k\}_{k \in [K]}$ between $e_q$ and $e_a$ in this query's context. We construct the personalized triple as $\{e_q, r_k, e_a\}_{k \in [K]}$ and denote this triple set as $\mathcal{H}(q, a, K)$. Then, the posterior distribution $Q(z)$ can be approximated, formally,

$$Q(z|q,a) \simeq \begin{cases} \frac{1}{K} & \text{if } z \in \mathcal{H}(q, a, K), \\ 0 & \text{else,} \end{cases} \tag{3}$$

where we assume a uniform distribution over the subset of personalized triples $\mathcal{H}(q, a, K)$. Therefore, the knowledge retrieval loss term can be calculated as

$$\mathcal{L}_{\text{retrieve}} = D_{KL}\left(Q(z)||P_{\theta,\mathcal{G}}\left(z|q\right)\right) = -\frac{1}{K}\sum_{z \in \mathcal{H}(q,a,K)} \log P_{\theta,\mathcal{G}}\left(z|q\right). \tag{4}$$

The detailed derivation can be found in Appendix A. By reducing $\mathcal{L}_{\text{retrieve}}$, we finetune the user's KG $\mathcal{G}$ such that the LLM can retrieve the personalized triples with higher confidence.

To collect $\mathcal{H}(q, a, K)$, users can either provide feedback about the relations between $e_q$ and $e_a$ in this query's context or extract the relations using the model $f_{\theta,\mathcal{G}}$. We design an instruction template to utilize the LLM to extract the relations $\{r_k\}_{k \in [K]}$ between $e_q$ and $e_a$:

> Based on the provided query and answer, identify K types of relationships between the subject
> <subject> and the object <object>, considering the context of the query and answer. <query>: $q$
> <answer>: $a$ <subject>: $e_q$ <object>: $e_a$ <relation>: [MASK][a]
>
> ___________
> [a]The LLMs in this paper are causal models that do not need a [MASK] token for generation. We include a [MASK] token within the templates in the paper to delineate the starting point for the model's generation, thereby aiding reader comprehension.

The instruction is fed into the LLM to generate $K$ relations formatted as

> $r_1$ <sep> $r_2$ <sep> ... <sep> $r_K$.

Then, we formulate $\mathcal{H}(q, a, K)$ as the set $\{e_q, r_k, e_a\}_{k \in [K]}$.

To calculate the retrieval probability $P_{\theta,\mathcal{G}}\left(z|q\right)$, we design an instruction template $\mathcal{T}_{\text{retrieve}}(\cdot)$ to instruct the LLM to predict what kind of the relationship is needed to answer this query:

> To answer the query: $q$, I need information $e_q$ [MASK]

We calculate the probability of a specific relation $r$ as $P_\theta \left( r | \mathcal{T}_{\text{retrieve}} \left( q \right) \right)$. Then, the retrieval probability of a knowledge triple $z = (e, r, e')$ is derived as

$$P_{\theta, \mathcal{G}} \left( z | q \right) = P_{\theta, \mathcal{G}} \left( (e, r, e') | q \right) = \begin{cases} \frac{P_\theta \left( r | \mathcal{T}_{\text{retrieve}}(q) \right)}{\sum_{z \in \mathcal{G}} P_{\theta, \mathcal{G}}(z|q)} & \text{if } e = e_q \text{ and } z \in \mathcal{G}, \\ 0 & \text{else.} \end{cases} \tag{5}$$

Thus, only the knowledge triples in $\mathcal{G}$ that start from $e_q$ have a retrieval probability larger than 0.

**Knowledge-enhanced Reasoning** The goal of the knowledge-enhanced reasoning term is to finetune the KG such that the retrieved personalized triples can most encourage the LLM to generate the correct answer. With the approximated posterior distribution $Q(z)$ in Equation (3), the reasoning loss term can be formulated as

$$\mathcal{L}_{\text{reasoning}} = -\mathbb{E}_{z \sim Q(z)} \left[ \log P_{\theta, \mathcal{G}}(a | q, z) \right] = -\frac{1}{K} \sum_{z \in \mathcal{H}(q, a, K)} \log P_{\theta, \mathcal{G}} \left( a | q, z \right). \tag{6}$$

We design an instruction template $\mathcal{T}_{\text{reasoning}}$ to instruct the model to predict the answer based on both the query $q$ and the knowledge triple $z$:

> Answer the query considering the user's personalized facts. \<question\>: $q$ \<facts\>: $z$ \<answer\>: [MASK]

The knowledge-enhanced reasoning probability is derived as $P_{\theta, \mathcal{G}} \left( a | q, z \right) = P_\theta \left( a | \mathcal{T}_{\text{reasoning}} \left( q, z \right) \right)$.

The final objective of KGT is the combination of knowledge retrieval optimization and knowledge-enhanced reasoning optimization, which is formulated as

$$\mathcal{L} = \mathcal{L}_{\text{retrieve}} + \mathcal{L}_{\text{reasoning}} = -\frac{1}{K} \sum_{z \in \mathcal{H}(q, a, K)} \log \left[ P_{\theta, \mathcal{G}} \left( a | q, z \right) P_{\theta, \mathcal{G}} \left( z | q \right) \right]. \tag{7}$$

### 4.3 OPTIMIZATION ALGORITHM

Unlike finetuning the model parameters with gradient descent, KGT is achieved by adding and removing the knowledge triples to and from the KG. For a triple $z$ and a KG $\mathcal{G}$, we have two operations: 1. $\mathcal{G} \oplus z \triangleq \mathcal{G} \cup \{z\}$; 2. $\mathcal{G} \ominus z \triangleq G \setminus \{z\}$. Because we focus on real-time model personalization, we formulate the optimization in the online learning setting where the optimization algorithm only has access to the user's current query $q_t$ at time $t$ and feedback $a_t$. The trivial solution to this query is that we remove all the triples in $\mathcal{G}$ that are probably retrieved based on $q_t$, which are the triples starting with $e_q$. Then we add the triple $z^*$ from $\mathcal{H}(q_t, a_t, K)$ with the highest reasoning probability $P_{\theta, \mathcal{G}} \left( a_t | q_t, z^* \right)$ into $\mathcal{G}$. In this case, $P_{\theta, \mathcal{G}} \left( z^* | q_t \right) = 1$ and the loss is minimized over $(q_t, a_t)$. However, this greedy solution will potentially hurt the other queries due to removing too much knowledge from the KG. Following the example in Section 4.1, the greedy solution will not only remove the triple $(Dog, Enjoy, Meat)$, but also remove other triples starting from the entity $Dog$ such as $(Dog, Is, Animal)$. The removed triples might be general or personalized factual knowledge that is essential to the other queries.

We propose a heuristic optimization algorithm based on two principles: 1. The computational cost should be low to achieve real-time personalization efficiency; 2. The KG should be less modified to preserve more knowledge. Both principles require us to modify as few triples as possible to achieve the optimization goal. The detailed algorithm is shown in Algorithm 1. In general, we evaluate the reasoning probability $P_{\theta, \mathcal{G}} \left( a_t | q_t, z' \right)$ for the triples $z'$ in $\mathcal{H}(q, a, K)$ and triples in $\mathcal{G}$ that starting from $e_{q_t}$. Then, we add the triples with the highest reasoning probability (i.e., the triples most encourage the LLM to generate the correct answer) from $\mathcal{H}(q, a, K)$ to $\mathcal{G}$ and remove the triples with the lowest reasoning probability from $\mathcal{G}$ iteratively until the model generates the correct answer.

---

**Algorithm 1** Knowledge Graph Tuning based on User Feedback

---

**Input:** Knowledge graph $\mathcal{G}$; KG-enhanced LLM $f_{\theta,\mathcal{G}}$; user's query $q_t$ at time $t$; user feedback $a_t$; size of personalized triple set for each query $K$; loss threshold $\epsilon$.
**Output:** Optimized KG $\mathcal{G}$.

1: Construct personalized triple set $\mathcal{H}(q_t, a_t, K)$
2: Extract the set of triples $\mathcal{G}_{q_t}$ in $\mathcal{G}$ starting from $e_{q_t}$.
3: $N_H \leftarrow |\mathcal{H}(q_t, a_t, K)|, N_{\mathcal{G}} \leftarrow |\mathcal{G}_{q_t}|$
4: $\{z_i^H, \lambda_i^H\}_{i \in [N_H]} \leftarrow \{z_i^H, P_{\theta,\mathcal{G}}(a_t|q_t, z_i^H)\}_{z_i^H \in \mathcal{H}(q_t, a_t, K)}$
5: $z_{1 \ldots N_H}^H \leftarrow z_{s(1) \ldots s(N_H)}^H$ where $s(i) = \mathrm{argsort}(\lambda_{1 \ldots N_H}^H, i)$
6: $\{z_i^{\mathcal{G}}, \lambda_i^{\mathcal{G}}\}_{i \in [N_{\mathcal{G}}]} \leftarrow \{z_i^{\mathcal{G}}, P_{\theta,\mathcal{G}}(a_t|q_t, z_i^{\mathcal{G}})\}_{z_i^{\mathcal{G}} \in \mathcal{G}_{q_t}}$
7: $z_{1 \ldots N_{\mathcal{G}}}^{\mathcal{G}} \leftarrow z_{s(1) \ldots s(N_{\mathcal{G}})}^{\mathcal{G}}$ where $s(i) = \mathrm{argsort}(\lambda_{1 \ldots N_{\mathcal{G}}}^{\mathcal{G}}, i)$
8: $\mathcal{L} \leftarrow -\frac{1}{K} \sum_{z \in \mathcal{H}(q_t, a_t, K)} \log\left[P_{\theta,\mathcal{G}}(a_t|q_t, z) P_{\theta,\mathcal{G}}(z|q_t)\right]$
9: $count\_add \leftarrow 0, count\_remove \leftarrow 0$
10: **while** $count\_add < N_H$ or $count\_remove < N_{\mathcal{G}}$ **do**
11:     **if** $count\_add < N_H$ **then**
12:         $count\_add \leftarrow count\_add + 1$
13:         $\mathcal{G} \leftarrow \mathcal{G} \cup z_{count\_add}^H$
14:         $\mathcal{L} \leftarrow -\frac{1}{K} \sum_{z \in \mathcal{H}(q_t, a_t, K)} \log\left[P_{\theta,\mathcal{G}}(a_t|q_t, z) P_{\theta,\mathcal{G}}(z|q_t)\right]$
15:         **if** $\mathcal{L} \leq \epsilon$ **then Break**
16:     **end if**
17:     **if** $count\_remove < N_{\mathcal{G}}$ **then**
18:         $count\_remove \leftarrow count\_remove + 1$
19:         $\mathcal{G} \leftarrow \mathcal{G} \setminus z_{count\_remove}^{\mathcal{G}}$
20:         $\mathcal{L} \leftarrow -\frac{1}{K} \sum_{z \in \mathcal{H}(q_t, a_t, K)} \log\left[P_{\theta,\mathcal{G}}(a_t|q_t, z) P_{\theta,\mathcal{G}}(z|q_t)\right]$
21:         **if** $\mathcal{L} \leq \epsilon$ **then Break**
22:     **end if**
23: **end while**
24: **return** $\mathcal{G}$

---

## 5 EXPERIMENTS

We evaluate KGT in terms of personalization performance and efficiency compared with the baselines. The experiments are conducted on a server with eight A100 GPUs.

### 5.1 EXPERIMENTAL SETUP

**Datasets** We first evaluate KGT in the setting where the user provides answers that conflict with the common factual knowledge that LLM learned from the pre-training dataset, ***which is an extremely hard setting of personalization.*** We evaluate KGT on two datasets: *CounterFact* (Meng et al., 2022) dataset and *CounterFactExtension* dataset we create based on *PARALLEL* dataset (Elazar et al., 2021) utilizing GPT-4 (OpenAI, 2024). The details about our dataset can be found in Appendix B. Both datasets consist of query-answer pairs involving factual knowledge that conflicts with reality.

Even though the counterfactual datasets can mimic the extremely hard personalization settings, to evaluate KGT in a more realistic setting, we build a real personalization interaction benchmark named ***PeInt***. We hired five real users to interact with the LLM (Llama3-8b) to discuss some topics based on their preferences and experience and build a benchmark containing five users' interactions. One example of interaction can be found in Appendix B. We collect all the interactions and evaluate if our method and the baselines can adapt to these personalized knowledge bases in the interactions.

To evaluate the real-time model personalization in practice, we sequentially input query-answer pairs into the model, ensuring that each pair is accessed only once during training. Once all the pairs in the dataset have been processed by the model, we assess the personalization effectiveness across the entire dataset.

**Baselines and Configurations** We compare KGT with fine-tunning (FT), ROME (Meng et al., 2022), Knowledge editing (KE) (De Cao et al., 2021), Knowledge neurons (KN) (Dai et al., 2022), and MEND (Mitchell et al., 2022). We also compare with relevant work, K-LaMP (Baek et al., 2024),

that aims to personalize the responses of LLMs with the knowledge from the knowledge store. For all baselines, we test several specifications of layers to edit and report the best results for each baseline. For FT, we follow previous arts (Meng et al., 2022) to execute full fine-tuning on a single layer. For KN, we specify a subset of all layers as candidates of knowledge neurons to reduce the search space for large models. And baseline results where no knowledge is edited are referred to as 'no edit.' We conduct experiments on GPT2-xl (Radford et al., 2019), Llama2-7B (Touvron et al., 2023), and Llama3-8B (AI, 2024). We equip each model with a KG, ConceptNet (Speer et al., 2017), to enhance the inference. $K$ in Equation (3) is set as five for experiments.

**Metrics** We utilize two metrics to evaluate the performance of personalization: (1) *Efficacy Score* (Meng et al., 2022), measuring the success rate of personalization using the training query-answer pairs directly. If the model generates the user's personalized answer with a higher probability than the answer before tuning, the model is considered successful for that pair. (2) *Paraphrase Score* (Meng et al., 2022), indicating the model's ability to accurately recall personalized knowledge in **paraphrased** forms. This assesses its personalization capacity while mitigating the impact of overfitting to specific contexts within the training dataset.

## 5.2 PERSONALIZATION PERFORMANCE

We evaluate the setting that the user only provides the answer $a$ as the feedback, and the model will extract the relations and construct $\mathcal{H}(q, a, K)$. The results on *CounterFact* dataset are shown in Table 1. It is shown that KGT outperforms the baselines significantly in both efficacy and paraphrase scores. Specifically, using Llama3-8B, KGT improves efficacy by more than 39%, 41%, 55%, 45%, 43%, 35%, and 61% on efficacy compared with FT, ROME, KE, KN, MEND, K-LaMP, and no-edit, respectively. For paraphrase score, KGT outperforms by more than 36%, 32%, 46%, 37%, 33%, 43%, 34% compared with FT, ROME, KE, KN, MEND, K-LaMP, and no-edit, respectively, on Llama3-8B. It is also observed that the results of KGT on Llama3 are better than Llama2. Our analysis is that Llama3 is more powerful in understanding and following instructions, which makes knowledge enhancement from the KG more effective. Such an improvement suggests that KGT will achieve better performance when the pre-trained LLMs are more and more powerful. Notably, K-LaMP achieves high efficacy but a low paraphrase score on GPT-2. K-LaMP focuses on recording entities rather than learning factual knowledge through interaction. As a result, it performs well when queries are repeated verbatim but struggles when queries are rephrased. Besides, simple entity records fail to contribute effectively to reasoning in more advanced models such as Llama2 and Llama3, leading to both low efficacy and paraphrase scores.

Table 1: Results on *CounterFact* dataset when the user only provides the answers to the queries as feedback.

| Method | GPT2 | | Llama2-7B | | Llama3-8B | |
|---|---|---|---|---|---|---|
| | Efficacy | Paraphrase | Efficacy | Paraphrase | Efficacy | Paraphrase |
| FT | $58.43\%_{\pm0.15\%}$ | $55.77\%_{\pm0.20\%}$ | $62.47\%_{\pm0.11\%}$ | $63.09\%_{\pm0.09\%}$ | $54.44\%_{\pm0.35\%}$ | $50.52\%_{\pm0.05\%}$ |
| ROME | $49.38\%_{\pm1.20\%}$ | $48.22\%_{\pm1.36\%}$ | $49.94\%_{\pm1.24\%}$ | $48.84\%_{\pm1.74\%}$ | $51.13\%_{\pm1.55\%}$ | $52.39\%_{\pm1.62\%}$ |
| KE | $51.50\%_{\pm0.32\%}$ | $51.85\%_{\pm0.27\%}$ | $34.25\%_{\pm1.63\%}$ | $30.45\%_{\pm1.43\%}$ | $40.56\%_{\pm1.21\%}$ | $41.00\%_{\pm0.57\%}$ |
| KN | $50.66\%_{\pm0.52\%}$ | $51.06\%_{\pm0.11\%}$ | $49.41\%_{\pm0.47\%}$ | $51.20\%_{\pm1.38\%}$ | $50.52\%_{\pm1.05\%}$ | $50.67\%_{\pm1.10\%}$ |
| MEND | $50.41\%_{\pm0.18\%}$ | $50.20\%_{\pm0.02\%}$ | $49.35\%_{\pm0.47\%}$ | $50.88\%_{\pm0.30\%}$ | $50.29\%_{\pm0.71\%}$ | $54.65\%_{\pm1.12\%}$ |
| K-LaMP | $96.17\%$ | $47.08\%$ | $65.73\%$ | $41.46\%$ | $59.41\%$ | $43.35\%$ |
| no edit | $35.87\%$ | $29.74\%$ | $30.58\%$ | $28.21\%$ | $33.52\%$ | $52.16\%$ |
| *KGT* | $91.77\%_{\pm1.37\%}$ | $91.75\%_{\pm1.84\%}$ | $91.1\%_{\pm1.43\%}$ | $83.86\%_{\pm1.03\%}$ | $94.58\%_{\pm0.96\%}$ | $86.89\%_{\pm1.37\%}$ |

KGT also shows significant improvement compared with the baselines on *CounterFactExtension*, and the results are shown in Table 2. Specifically, KGT improves efficacy by more than 31%, 34%, 51%, 42%, 48%, 57%, and 54% on efficacy when adopting Llama3-8B compared with FT, ROME, KE, KN, MEND, K-LaMP, and no-edit, respectively. For paraphrase score on Llama3-8B, KGT outperforms by more than 27%, 35%, 41%, 36%, 44%, 29%, and 42% compared with FT, ROME, KE, KN, MEND, K-LaMP, and no-edit, respectively.

KGT also shows significant improvement over the baselines in realistic personalization setting on *PeInt*. The results on *PeInt* are shown in Table 3. Specifically, KGT improves efficacy by more than 71%, 76%, 75%, 79%, 77%, 61% and 75% on efficacy when adopting Llama3-8B compared with FT, ROME, KE, KN, MEND, K-LaMP, and no-edit, respectively. For paraphrase score on Llama3-8B,

Table 2: Results on *CounterFactExtend* dataset when the user only provides the answers to the queries as feedback.

| Method | GPT2 | | Llama2-7B | | Llama3-8B | |
|---|---|---|---|---|---|---|
| | Efficacy | Paraphrase | Efficacy | Paraphrase | Efficacy | Paraphrase |
| FT | $58.67\%_{\pm0.11\%}$ | $53.71\%_{\pm0.06\%}$ | $59.70\%_{\pm0.15\%}$ | $63.09\%_{\pm0.09\%}$ | $62.29\%_{\pm0.25\%}$ | $61.97\%_{\pm0.10\%}$ |
| ROME | $57.44\%_{\pm1.75\%}$ | $58.45\%_{\pm1.00\%}$ | $47.33\%_{\pm1.60\%}$ | $48.36\%_{\pm0.71\%}$ | $59.28\%_{\pm1.79\%}$ | $53.77\%_{\pm1.37\%}$ |
| KE | $52.49\%_{\pm0.28\%}$ | $52.55\%_{\pm0.44\%}$ | $33.83\%_{\pm1.91\%}$ | $44.49\%_{\pm0.92\%}$ | $41.92\%_{\pm1.16\%}$ | $47.35\%_{\pm0.47\%}$ |
| KN | $47.40\%_{\pm0.37\%}$ | $47.22\%_{\pm0.04\%}$ | $49.74\%_{\pm0.05\%}$ | $49.67\%_{\pm1.36\%}$ | $51.51\%_{\pm0.58\%}$ | $52.62\%_{\pm1.91\%}$ |
| MEND | $58.30\%_{\pm0.12\%}$ | $58.51\%_{\pm0.07\%}$ | $40.73\%_{\pm0.05\%}$ | $43.61\%_{\pm0.04\%}$ | $45.62\%_{\pm0.10\%}$ | $44.34\%_{\pm1.75\%}$ |
| K-LaMP | $84.77\%$ | $52.66\%$ | $74.73\%$ | $46.08\%$ | $36.60\%$ | $59.44\%$ |
| no edit | $47.14\%$ | $51.74\%$ | $30.22\%$ | $42.93\%$ | $39.26\%$ | $47.04\%$ |
| *KGT* | $82.57_{\pm2.82\%}$ | $78.35\%_{\pm3.26\%}$ | $90.68_{\pm0.74\%}$ | $83.8_{\pm1.20\%}$ | $93.80\%_{\pm0.36\%}$ | $89.22\%_{\pm1.17\%}$ |

KGT outperforms by more than 69%, 62%, 63%, 67%, 71%, 79%, and 63% compared with FT, ROME, KE, KN, MEND, K-LaMP, and no-edit, respectively.

Table 3: Results on *PeInt* dataset when the user only provides the answers to the queries as feedback.

| Method | GPT2 | | Llama2-7B | | Llama3-8B | |
|---|---|---|---|---|---|---|
| | Efficacy | Paraphrase | Efficacy | Paraphrase | Efficacy | Paraphrase |
| FT | $26.53\%_{\pm0.00\%}$ | $16.32\%_{\pm0.00\%}$ | $30.61\%_{\pm0.02\%}$ | $20.40\%_{\pm0.01\%}$ | $24.48\%_{\pm0.00\%}$ | $20.41\%_{\pm0.00\%}$ |
| ROME | $17.68\%_{\pm3.11\%}$ | $14.96\%_{\pm2.36\%}$ | $22.44\%_{\pm0.00\%}$ | $17.00\%_{\pm1.17\%}$ | $19.04\%_{\pm1.17\%}$ | $27.21\%_{\pm1.17\%}$ |
| KE | $77.55\%_{\pm0.00\%}$ | $75.51\%_{\pm0.00\%}$ | $17.68\%_{\pm1.17\%}$ | $18.36\%_{\pm0.00\%}$ | $20.40\%_{\pm0.02\%}$ | $26.53\%_{\pm0.02\%}$ |
| KN | $12.24\%_{\pm0.00\%}$ | $6.12\%_{\pm0.00\%}$ | $18.36\%_{\pm0.01\%}$ | $16.32\%_{\pm0.00\%}$ | $16.32\%_{\pm0.00\%}$ | $22.45\%_{\pm0.01\%}$ |
| MEND | $10.88\%_{\pm1.17\%}$ | $11.56\%_{\pm1.17\%}$ | $67.34\%_{\pm0.07\%}$ | $65.30\%_{\pm0.03\%}$ | $18.36\%_{\pm0.01\%}$ | $18.36\%_{\pm0.00\%}$ |
| K-LaMP | $12.24\%$ | $20.40\%$ | $16.32\%$ | $24.48\%$ | $34.69\%$ | $10.20\%$ |
| no edit | $12.24\%$ | $6.12\%$ | $18.37\%$ | $18.36\%$ | $20.40\%$ | $26.53\%$ |
| *KGT* | $89.41\%_{\pm0.25\%}$ | $82.31\%_{\pm0.75\%}$ | $93.78\%_{\pm1.03\%}$ | $83.75\%_{\pm0.82\%}$ | $95.75\%_{\pm0.68\%}$ | $89.83\%_{\pm0.37\%}$ |

The improvement in both efficacy and paraphrase rates demonstrates that KGT outperforms the baseline in personalization performance significantly.

## 5.3 EFFICIENCY COMPARISON

In addition to the personalization performance, we also evaluate KGT's efficiency in terms of latency and GPU memory cost. The method must achieve low latency and low GPU memory cost to realize **real-time personalization** in practice under the setting with limited computational resources. KGT can improve the time and GPU memory cost efficiency because it only requires inference, which is far more efficient than back-propagation. For latency, we test the average time that our method and the baselines require to complete the personalization for one query-answer pair. The latency results on *CounterFact* data are shown in Table 4. The results demonstrate that KGT achieves significant latency reduction compared with most baselines. Notably, Llama3 requires less time on several baselines than Llama2 and GPT2 because we stop training once the loss converges. The results on the GPU memory cost can also be found in Table 4. The memory cost is reduced significantly because only the inference is required for KGT. Specifically, KGT reduces 57%, 56%, 77%, 63%, 62% and 52% GPU memory cost when adopting Llama3-8B compared with FT, KE, KN, MEND, and K-LaMP, respectively.

## 5.4 ABLATION STUDY

### 5.4.1 DOES USER NEED TO PROVIDE FEEDBACK OF RELATIONS?

We also conduct experiments under the setting where the user also provides feedback on relations $r$ between $e_q$ and $e_a$ under the query's context. We utilize GPT-4 to mimic the user and extract the relations to construct $\mathcal{H}(q, a, K)$. We use the same instruction template in Section 4.2 for the GPT-4 to conduct relation extraction. The compared results with and without the user's feedback of relations are shown in Figure 3. It is shown that KGT achieves similar performance if the user provides feedback on the relations in addition to the answer. Notably, utilizing the relations extracted by the LLM $\theta$ can even achieve higher performance on efficacy and paraphrase rates in most cases. Our analysis is that when extracting relations using the LLM $\theta$, KGT implicitly distills knowledge

Table 4: Results of latency and GPU memory costs on *CounterFact* dataset when the user only provides the answers to the queries as feedback.

| Method | GPT2 | | Llama2-7B | | Llama3-8B | |
|---|---|---|---|---|---|---|
| | Memory | Latency | Memory | Latency | Memory | Latency |
| FT | 8516MB | 1.80s | 30990MB | 0.81s | 36968MB | 0.25s |
| ROME | 11948MB | 1.39s | 30452MB | 2.33s | 36660MB | 2.05s |
| KE | 31574MB | 2.18s | 33464MB | 0.30s | 69542MB | 0.13s |
| KN | 12832MB | 3.55s | 56148MB | 0.69s | 44000MB | 0.34s |
| MEND | 11036MB | 0.86s | 35166MB | 1.98s | 42428MB | 1.40s |
| K-LaMP | 9532MB | 0.03s | 28204MB | 0.09s | 33724MB | 0.06s |
| *KGT* | 6686MB | 0.16s | 13516MB | 0.14s | 15904MB | 0.15s |

from the model to the knowledge triples, which might benefit the model inference more than human feedback on relations. Thus, in practice, the user will only need to provide the personalized answer to a query as feedback to KGT.

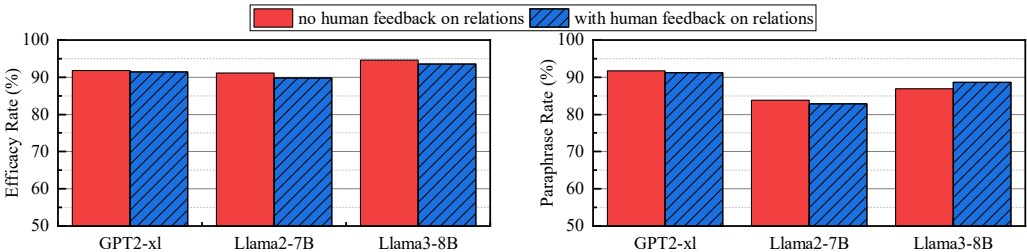

Figure 3: Compared results on *CounterFact* dataset with and without user feedback on relations.

### 5.4.2 EFFECT OF THE SIZE OF QUERY SET

We also evaluate the effect of the query set size on the personalization performance. We conduct experiments on the $CounterFact$ dataset with the Llama3-8B model and evaluate KGT and baselines with query sets of different sizes. The results are shown in Figure 4. It is shown that when the size of the query set increases, the performance of baselines degrades dramatically, while KGT can preserve high performance. This compared result illustrates that KGT can be scaled to a large amount of personalized knowledge. This scalability is crucial for meeting the long-term needs of users who require the accumulation of extensive personalized knowledge when using LLMs.

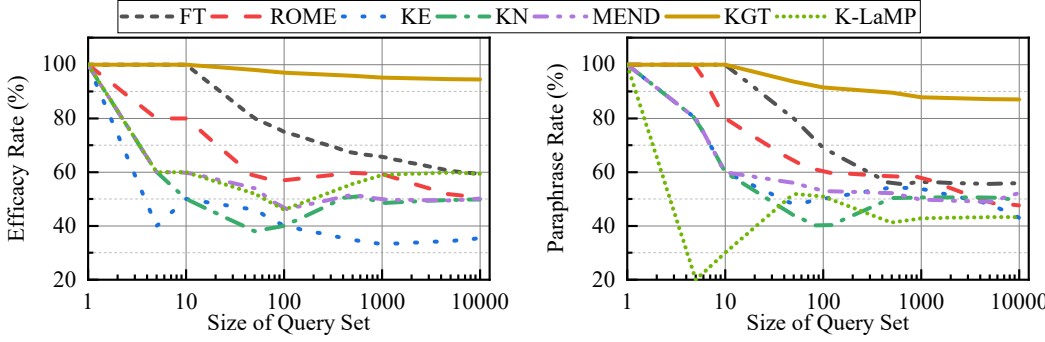

Figure 4: Compared results on *CounterFact* dataset using Llama3-8B with different query set sizes.

## 6 CONCLUSION AND LIMITATION

We propose an approach, KGT, that personalizes models by optimizing external knowledge graph rather than model parameters. Our method and experimental results demonstrate that our approach offers benefits in terms of performance and efficiency, as supported by empirical studies. By addressing the critical challenges of efficiency and interpretability in model personalization, KGT offers a promising direction for future research and application in enhancing user interactions with LLMs, which has a positive societal impact. For this work, we only focus on the single-user scenario. When there are multiple users, we can maintain just one base LLM and multiple personalized KGs rather than multiple LLMs, which is more storage-efficient. When serving inferences for multiple users, since there is only one base model, we can package their queries with the corresponding knowledge triples in one batch and conduct inferences in parallel. Thus, KGT can achieve high efficiency when serving multiple users. One limitation of this method (KGT) is that it depends on the LLM's ability to follow instructions when calculating $P_{\theta,\mathcal{G}}(a|q,z)$, $P_{\theta,\mathcal{G}}(z|q)$, and collecting $\mathcal{H}(q,a,K)$. However, the existing state-of-the-art LLMs already meet KGT's requirements for this capability, and future LLMs will possess even stronger abilities to follow instructions.

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

# A  DERIVATION OF $\mathcal{L}_{\text{RETRIEVE}}$

We provide the derivation of the retrieve loss. By approximating the $Q(z)$ with $Q(z|q,a)$, the KL divergence is calculated as

$$
\begin{aligned}
\mathcal{L}_{\text{retrieve}} = D_{KL}\left(Q(z)||P_{\theta,\mathcal{G}}\left(z|q\right)\right) &= D_{KL}\left(Q(z|q,a)||P_{\theta,\mathcal{G}}\left(z|q\right)\right) \\
&= \mathbb{E}_{z\sim Q(z|q,a)}\left[\log Q(z|q,a) - \log P_{\theta,\mathcal{G}}(z|q)\right] \\
&= -\mathbb{E}_{z\sim Q(z|q,a)}\log P_{\theta,\mathcal{G}}(z|q) + \mathbb{E}_{z\sim Q(z|q,a)}\log Q(z|q,a) \\
&= -\mathbb{E}_{z\sim Q(z|q,a)}\log P_{\theta,\mathcal{G}}(z|q) + \text{CONST},
\end{aligned}
\tag{8}
$$

where the term $\mathbb{E}_{z\sim Q(z|q,a)}\log Q(z|q,a)$ is not related to $\mathcal{G}$ and can be treated as a constant. With $Q(z|q,a)$ formulated in Equation (3), the loss can be calculated as

$$
\begin{aligned}
\mathcal{L}_{\text{retrieve}} &= -\mathbb{E}_{z\sim Q(z|q,a)}\log P_{\theta,\mathcal{G}}(z|q) + \text{CONST} \\
&= \sum_{z\in\mathbb{Z}} Q(z|q,a)\log P_{\theta,\mathcal{G}}\left(z|q\right) + \text{CONST} \\
&= -\frac{1}{K}\sum_{z\in\mathcal{H}(q,a,K)}\log P_{\theta,\mathcal{G}}\left(z|q\right) + \text{CONST},
\end{aligned}
\tag{9}
$$

where the CONST is omitted in the optimization.

# B  DETAILS ON THE *CounterFactExtension* DATASET

*CounterFactExtension* is a dataset that extends the *CounterFact* with 27737 additional samples. Specifically, *CounterFactExtension* is on the task of knowledge editing for *Efficacy* and *Paraphrase* evaluation. Likewise, *CounterFactExtension* is crafted to identify the differences between minor adjustments in the vocabulary used by the model and substantial, widespread shifts in the foundational factual knowledge.

*CounterFactExtension* is built on knowledge pairs processed by Dai et al. (Dai et al., 2022) based on PARAREL dataset (Elazar et al., 2021). Specifically, for each knowledge pair, we employ GPT-4 to extract the subject, relation, and the true target, $t = (s, r, o^*)$, and generate a counterfactual target, $o^c$. Further, a generation prompt (generated by GPT-4) is also provided for the *Paraphrase* evaluation. Finally, we manually check each sample and the generation to ensure the correctness.

## B.1  DATASET SAMPLES

```
 1      {
 2          "case_id": 0,
 3          "requested_rewrite": {
 4              "prompt": "{} works in the field of ",
 5              "relation_id": "P101(field of work)",
 6              "target_new": {
 7                  "str": "mechanical engineering",
 8              },
 9              "target_true": {
10                  "str": "logic",
11              },
12              "subject": "Alan Turing"
13          },
14          "generation_prompts": [
15              "The field that Alan Turing is associated with is"
16          ]
17      },
```

```
{
    "case_id": 25585,
    "requested_rewrite": {
        "prompt": "{} maintains diplomatic relations with ",
        "relation_id": "P530(diplomatic relation)",
        "target_new": {
            "str": "Umboria",
        },
        "target_true": {
            "str": "Malaysia",
        },
        "subject": "Ukraine"
    },
    "generation_prompts": [
        "Diplomatic relations are established between Ukraine
            and"
    ]
},
```

one example of interactions in our constructed **_PeInt_** bench mark is shown below:

> User 1: "Can you update me on the latest publications related to quantum entanglement from the past month and also help me plan my day tomorrow? I have a full day at the lab."
>
> AI Assistant: "I found five new publications that match your criteria, including two from the Quantum Information Journal. Would you like a summary of each, or should I focus on those with experimental results similar to your project? Also, I see you have a busy day at the lab tomorrow. There's flexibility in your afternoon schedule; how about scheduling some violin practice for the evening?"
>
> User 1: "Please summarize the ones with experimental results, and download the full papers for later. Schedule violin practice at 7:00 PM and remind me to take Schrödinger for his vet appointment."
>
> AI Assistant: "I've scheduled violin practice for 7:00 PM and set a reminder for Schrödinger's vet appointment at 3:00 PM, with a 30-minute buffer for travel. The summaries of the selected publications will be ready in your inbox within the hour. Is there anything else I can assist with?"
>
> User 1: "Thanks! Could you also check for any chess tournaments happening this weekend? I'd like to participate if possible."
>
> AI Assistant: "I will look into the local chess club listings and let you know about the tournaments this weekend. Anything else you need?"

There are several personal factual knowledge contained in this example, including that user 1's research field is related to quantum physics, user 1's pet is called Schrödinger, and user 1's hobby is chess. We collect all these interactions and evaluate if our method and the baselines can adapt to these personalized knowledge bases.

