# OpenReview forum: "Knowledge Graph Tuning: Real-time Large Language Model Personalization based on Human Feedback"
_ICLR.cc/2025/Conference — Submitted to ICLR 2025_

### Official Review · Reviewer_N6xa · 2024-10-28

**Soundness:** 3
**Presentation:** 3
**Contribution:** 2
**Rating:** 6
**Confidence:** 3

**Summary:**

This paper introduces a novel method called Knowledge Graph Tuning (KGT) to personalize large language models (LLMs) in real time. Unlike traditional methods that rely on computationally expensive and memory-intensive back-propagation, KGT leverages external knowledge graphs (KGs) to manage user-specific information. This enables the model to adjust without altering its core parameters, improving efficiency and interpretability. Experimental results demonstrate that KGT significantly enhances personalization performance while lowering latency and resource requirements, showing promise in delivering efficient, effective, and interpretable real-time personalization.

**Strengths:**

The strengths of the paper lie primarily in its innovative approach to personalization for LLMs. By avoiding the use of back-propagation, KGT significantly reduces computational overhead, leading to faster processing times and lower resource consumption. This makes it particularly suitable for real-time applications, where latency can be a critical factor. Additionally, KGT offers a notable advantage in interpretability. By managing user-specific knowledge through adjustments to external knowledge graphs rather than internal parameters, the process becomes more transparent. This makes it easier to trace how user interactions influence changes, allowing for clearer insights into the personalization mechanics. The scalability of KGT is also a strong point, as the method handles an increasing volume of queries effectively, which is essential for users who accumulate extensive personalized knowledge over time. Experimental results back up these claims, showing that KGT performs better than existing methods in personalization benchmarks.

**Weaknesses:**

The success of KGT heavily depends on the quality and completeness of the knowledge graphs it uses. If these graphs contain inaccuracies or gaps, the effectiveness of personalization can be compromised, limiting the model's ability to provide relevant responses. Another limitation lies in its reliance on the LLM’s ability to follow instructions accurately when interpreting the knowledge graph. This could lead to inconsistent results if deployed on language models with varied instruction-following capabilities. Furthermore, while KGT aims to reduce user effort by handling knowledge updates through interactions, there are cases where explicit user feedback might still be required. This adds an element of manual intervention, potentially diminishing the appeal of fully automated personalization.

**Questions:**

Can the authors elaborate on how KGT handles contradictory user feedback over time, especially when a user’s personalized knowledge evolves and conflicts with previously stored information in the knowledge graph?

---

> ### Author Response · Authors · 2024-11-20
> **Rebuttal**
>
> Thanks for your review and constructive suggestions. We are happy to address your concerns.
>
> **Reliance on the quality and completeness of the knowledge graphs.**
>
> Instead of heavily relying on the quality of KGs, our algorithm is more of a method to improve the quality and completeness of the knowledge graph during the daily use of LLMs. Even if the KG is low-quality and incomplete, our method can inject and update the knowledge triples in the graph automatically based on the user’s interaction with the LLM. In addition, we want to emphasize that with the development of LLM, people realize that the structural knowledge base, such as knowledge graphs, is valuable to enhance LLM's reasoning. More and more knowledge graphs are developed and proven effective in cooperating with LLMs[1-3], including commonsense knowledge graphs[4] and some domain-specific graphs like Medical Knowledge Graphs (MKG)[5,6]. We believe that the quality and quantity of KG will improve continually, especially with the rapid development of LLMs and knowledge-enhanced inference technologies.
>
> **Reliance on the LLM’s ability to follow instructions.**
>
> It is true that our method, along with all the other retrieval-based methods, relies on the LLM’s ability to follow instructions. However, it is notable that there is a clear tendency for advanced pre-trained LLMs to have a stronger ability to follow instructions since these models are to be fine-tuned using Reinforcement Learning from Human Feedback (RLHF) [7]. Thus, our method is scalable to more advanced LLMs.
>
> **User effort.**
>
> Our algorithm does not require the user to provide an “explicit” and "correct" answer.  All we need is to extract the personalized triples containing factual knowledge from the AI-user interactions. For example, when you use video platforms or shop online, you input some information or choose some entities/recommendations that contain your preference. These interactions contain personalized factual knowledge that our algorithm can utilize to edit the user's KG and realize personalization. Our algorithm does not require the user to provide any information in addition to normal interactions while employing LLM to extract necessary knowledge. A practical example of such interaction is shown in appendix B.1.
>
> **Conflicts with previously stored information in the knowledge graph.**
>
> Thanks for your insightful question, and this is what we intended to demonstrate via the evaluations on CounterFact datasets. For the CounterFact datasets, all the feedback from the user is contradictory to the knowledge graph we employed. The performance on such datasets shows the effectiveness of updating KGs using our algorithm.
>
>
>
>
> [1] Ting-Yun Chang, Yang Liu, Karthik Gopalakrishnan, Behnam Hedayatnia, Pei Zhou, and Dilek Hakkani-Tur. Incorporating commonsense knowledge graph in pretrained models for social commonsense tasks. arXiv preprint arXiv:2105.05457, 2021.
>
> [2] Yichong Xu, Chenguang Zhu, Shuohang Wang, Siqi Sun, Hao Cheng, Xiaodong Liu, Jianfeng Gao, Pengcheng He, Michael Zeng, and Xuedong Huang. Human parity on commonsenseqa: Augmenting self-attention with external attention. arXiv preprint arXiv:2112.03254, 2021.
>
> [3] Yunzhi Yao, Shaohan Huang, Li Dong, Furu Wei, Huajun Chen, and Ningyu Zhang. Kformer: Knowledge injection in transformer feed-forward layers. In CCF International Conference on Natural Language Processing and Chinese Computing, pages 131–143. Springer, 2022.
>
> [4] Speer, Robyn, Joshua Chin, and Catherine Havasi. "Conceptnet 5.5: An open multilingual graph of general knowledge." Proceedings of the AAAI conference on artificial intelligence. Vol. 31. No. 1. 2017.
>
> [5] Wu, Xuehong, et al. "Medical knowledge graph: Data sources, construction, reasoning, and applications." Big Data Mining and Analytics 6.2 (2023): 201-217.
>
> [6] Gong, Fan, et al. "SMR: medical knowledge graph embedding for safe medicine recommendation." Big Data Research 23 (2021): 100174.
>
> [7] Ouyang, L., Wu, J., Jiang, X., et al. (2022). “Training language models to follow instructions with human feedback.”
>
> [8] Lewis, Patrick, et al. Retrieval-augmented generation for knowledge-intensive nlp tasks. Advances in Neural Information Processing Systems 33 (2020): 9459-9474.

---

> > ### Comment · Reviewer_N6xa · 2024-11-25
> >
> > Thank you for your detailed response. After considering your clarifications, I have a few additional thoughts:
> >
> > 1. Knowledge Graph Quality:
> > Your approach of improving the knowledge graph over time is valuable, but more details are needed on how KGT ensures the accuracy of newly injected triples, particularly in ambiguous or complex domains (e.g., medical, legal). How do you handle errors or contradictions in user-provided data?
> >
> > 2. Instruction Following and Robustness:
> > While advanced LLMs fine-tuned with RLHF tend to follow instructions better, what happens when KGT is applied to less advanced models or those with inconsistent instruction-following abilities? A clearer analysis of how KGT scales with various LLMs would be helpful.
> >
> > 3. User Effort and Feedback:
> > The lack of explicit user feedback is a strong point, but I’m still curious how KGT handles ambiguous or conflicting interactions (e.g., a user changing preferences over time). Can you provide more details on how KGT manages contradictions in real-world scenarios?
> >
> > 4. Scalability and Efficiency:
> > Your method’s scalability is promising, but further benchmarks would help clarify how KGT performs as the knowledge graph grows or as the system handles more users. Does it maintain low latency and resource efficiency at scale?

---

> > > ### Author Response · Authors · 2024-11-26
> > > **Response to Reviewer N6xa**
> > >
> > > Thanks for your reply. We are delighted to answer your questions.
> > >
> > > 1.	We followed the previous works’ settings [7,8] to realize the personalization and editing based on the user-LLM interactions. While there might be some errors in user-provided data in the real world, it is not our work’s focus to filter the user’s data. However, it is notable that our method can remind the user of the knowledge triples that are to be removed and added, which provides a more interpretable way of editing and controlling the LLM system. Since filtering the user’s error data is not our focus, then for the contradictions in user-provided data, our work would treat them as changing the user’s preference and editing the KG to align with the latest user data.
> > >
> > > 2.	Thanks for your constructive advice. We conducted experiments on GPT2, Llama2, and Llama3 in our work. It showed that the performance of KGT will scale up when the model gets more advanced. We understand your concern about the less advanced models. Still, the performance of these models will be manageable as long as they have the in-context learning capability (which is a common capability of LLMs). The performance can also be boosted by adapting instruction tuning to make the model follow our template of injecting knowledge triples and prompts.
> > >
> > > 3.	When the user changes preferences over time, KGT will remove the stale knowledge triples that will lead to the “wrong” response and add the new knowledge triples into the knowledge graph. For example, a user majored in physics and now changes to computer science. Then, after the user provides the feedback as something like “Oh, I now major in computer science,” KGT will detect the stale knowledge triple <user; major; physics> and replace it with the KGT-formulated knowledge triple < user; major; computer_science>.
> > >
> > > 4.	Thanks for your insightful thoughts. The results in section 5.4.2 show the scalability of KGT under a larger query-answer volume (which is equivalent to the larger graph), and the results of Table 4 are the average results under the large query-answer set. For this work, we only focus on the single-user scenario, but we are delighted to discuss the multi-user setting. When there are multiple users, we can maintain just one base LLM and multiple personalized KGs rather than multiple LLMs, which is more storage-efficient. When serving inferences for multiple users, since there is only one base model, we can package their queries with the corresponding knowledge triples in one batch and conduct inferences in parallel. Thus, KGT can achieve high efficiency when serving multiple users. Thanks again for your suggestions, and we have included this discussion in the draft.

---

### Official Review · Reviewer_kufp · 2024-10-30

**Soundness:** 2
**Presentation:** 3
**Contribution:** 2
**Rating:** 3
**Confidence:** 4

**Summary:**

This paper proposes a novel approach called Knowledge Graph Tuning (KGT) for real-time large language model personalization based on user feedback. The paper addresses the challenges of computational efficiency and interpretability in model personalization by leveraging knowledge graphs (KGs) to optimize the KGs instead of the model parameters. The proposed method is evaluated on state-of-the-art LLMs and compared with several baselines, showing significant improvements in personalization performance while reducing latency and GPU memory costs.

**Strengths:**

(1) The paper tackles a significant issue in large language models: real-time model personalization based on user feedback. The proposed method presents a promising solution to improve user experience by tailoring LLMs to individual knowledge.

(2) The incorporation of knowledge graphs (KGs) in the proposed method is both innovative and practical. The author offers a theoretical analysis of the method's design.

(3) The experimental evaluation is thorough, featuring tests on multiple datasets with state-of-the-art LLMs. The results highlight the effectiveness and efficiency of the proposed KGT method compared to various baselines.

**Weaknesses:**

(1) The novelty of the proposed method is limited. The proposed framework is a typical retrieval-based method that relies on including retrieved facts in the context to tailor the output of LLMs. Similar ideas have been proposed by previous studies [1] and even have been applied in ChatGPT today with the proposed "memory module" (similar open-source version here: https://github.com/mem0ai/mem0).

(2) The proposed method relies on explicit human feedback to "optimize" the retrieval process, which largely limits its practicability. Moreover, it relies on structured KGs, which are hard to construct. Nowadays, ChatGPT and Mem0 can unsupervised and automatically extract memory from conversational contexts for personalized outputs, which is more flexible.

Although the author proposes a theoretical analysis to explain the "optimization of retrieval". The analysis is trivial, which is widely adopted in the RAG such as [2]

(3) The effectiveness of the proposed method is only evaluated on the fact editing tasks. However, other aspects of alignment are not discussed in the paper, such as HarmfulQ, and compared with other efficient alignment methods, such as Deal [3].

[1] Baek, J., Chandrasekaran, N., Cucerzan, S., Herring, A., & Jauhar, S. K. (2024, May). Knowledge-augmented large language models for personalized contextual query suggestion. In Proceedings of the ACM on Web Conference 2024 (pp. 3355-3366).
[2] LUO, L., Li, Y. F., Haf, R., & Pan, S. Reasoning on Graphs: Faithful and Interpretable Large Language Model Reasoning. In The Twelfth International Conference on Learning Representations.
[3] Huang, J. Y., Sengupta, S., Bonadiman, D., Lai, Y. A., Gupta, A., Pappas, N., ... & Roth, D. (2024). Deal: Decoding-time alignment for large language models. ICML 2024.

**Questions:**

(1) Can the proposed method generalize to other alignment tasks and compare the performance with other efficient alignment methods?
(2) Can you compare it with other LLM personalized methods instead of just knowledge editing ones?

---

> ### Author Response · Authors · 2024-11-20
> **Rebuttal**
>
> Thanks for your review and constructive suggestions. We would like to respond to your concerns.
> **Weakness1.**
> It is true that retrieval-based methods have been explored intensively. However, we want to claim that the main contribution and novelty of our paper is to maintain and update the knowledge base rather than simply retrieving and reasoning. It is trivial to collect personal context through conversations and conduct retrieval-based reasoning based on that knowledge base nowadays, but it is not trivial to detect conflicting knowledge facts during the LLM-AI interaction and update the stale knowledge in the knowledge base, which was not explored by the reference you provided and is the focus of our paper. We still appreciate the reference you provided and have included it and the discussion in our revised draft’s related works.
>
> **Weakness2.**
> We would like to address your concern about human effort first. Our algorithm does not require the user to provide an “explicit” and "correct" answer.  All we need is to extract the personalized triples containing factual knowledge from the AI-user interactions. For example, when you use video platforms or shop online, you input some information or choose some entities/recommendations that contain your preference. These interactions contain personalized factual knowledge that our algorithm can utilize to edit the user's KG and realize personalization. We do not require the user to provide any information in addition to normal interactions. We employ LLM to extract that necessary knowledge, which can “unsupervised and automatically extract memory from conversational contexts for personalized outputs” as you claimed. A practical example of such interaction is shown in appendix B.1.
> Regarding your concern about constructing KGs, we would like to emphasize that with the development of LLM, people realize that the structural knowledge base, such as knowledge graphs, is valuable to enhance LLM's reasoning. More and more knowledge graphs are developed and proven effective in cooperating with LLMs[1-3], including commonsense knowledge graphs[4] and some domain-specific graphs like Medical Knowledge Graphs (MKG)[5,6]. We believe that the quality and quantity of KG will improve continually, especially with the rapid development of LLMs and knowledge-enhanced inference technologies. Thus, we think it is a promising research direction to apply KG to LLM reasoning. [References sees reply for reviewer 5.]
> Our theoretical analysis is inspired by RAG[8]. However, this is the first work that conducts optimization on the knowledge base rather than only retrieving and reasoning, which is novel. Thanks for providing the reference; we have included it in our revised draft. [References sees reply for reviewer 5.]
>
> **Weakness 3 and Questions 1 & 2.**
> We do not only evaluate the fact editing tasks. To evaluate KGT in a more realistic setting, we build and evaluate on a real personalization interaction benchmark PeInt in our paper. Thanks for your suggestion to evaluate the alignment tasks. We want to emphasize that our method focuses on the deployment phase, which is after the RLHF step, as shown in Figure 1. Our method focuses on the penalization that is realized by editing the specific factual knowledge rather than aligning the LLM to some general preference. Even though our paper does not focus on the alignment, we are still delighted to conduct experiments and show the results that are compared with the reference (DeAL) you provided. We conduct experiments on the HarmfulQ dataset with the MPT-7B model, which is not aligned with RLHF. The compared results are shown below:
>
> | Method       | harmless ↑    |
> |--------------|---------------|
> | base model   | 35%           |
> | DeAL         | 100%          |
> | KGT          | 99.5%         |
>
> It is shown that even though KGT does not focus on alignment tasks, it can also align the LLM to reduce harmfulness by indicating some harmful factual knowledge in the KG. Besides knowledge editing methods, we also compare KGT with K-LaMP (Baek et al., 2024) in our paper, which aims to personalize the responses of LLMs with the knowledge from the knowledge store.

---

> > ### Comment · Reviewer_kufp · 2024-11-20
> >
> > Thanks the author for the response. However, it might not well address my concerns. The reasons and following questions are below.
> >
> > 1.**Explicit user feedback**: In Lines 232-240, the author states
> > > The goal of the knowledge retrieval term is to finetune the KG such that the LLM can retrieve personalized triples based on the **user’s feedback**. Given a query $q$ and the user’s feedback $a$ ...
> >
> > If no user's feedback is given, how to obtain the $a$? How do we extract the personalized triples for optimization as they require feedback stated in Lines 236-237.
> >
> > 2. **Information Leakage**: As shown in Algorithm 1 and the description in section 4.3, the "optimization" is to maximize the probability of generating the correct answer.  $a$. In the experiments, the answer $a$ is directly defined as the desired user's answer. Then, as stated in Lines 318-319
> > > remove the triples with the lowest reasoning probability from G iteratively until the model generates the correct answer.
> >
> > Would the ground-truth answer be the most strong explicit user's feedback? Would this process lead to information leakage issues as it is directly "optimized" with ground truth?
> >
> > 3. **Novelty**: The proposed method is just finding facts from KGs and put into context to modify LLM outputs. Similar RAG-based have been proposed in recent works, even with the usage of KGs [1,2].
> >
> > [1] Retrieval-enhanced Knowledge Editing in Language Models for Multi-Hop Question Answering, CIKM 2024
> > [2] MQUAKE  Assessing Knowledge Editing in language Models via Multi Hop Questions, EMNLP 2023
> >
> > 4. **Evaluation Metrics**: The proposed method is evaluated on some knowledge editing datasets. However, it does not report the edit accuracy that is used in compared baselines. It is hard to justify the effectiveness of the proposed method. Nevertheless, I would imagine a 100% edit accuracy under the current optimization objects. How can we evaluate the generalizability of the proposed method? Can the personalization generalize to other questions?

---

> > > ### Author Response · Authors · 2024-11-20
> > > **Response to Reviewer kufp**
> > >
> > > Thanks for your response! We are more than happy to respond to your concerns further.
> > >
> > > **Explicit user feedback**
> > >
> > > We still require user feedback, but not “explicit” and “correct” feedback, as we claimed in our initial response. The feedback can just be the normal conversations between the user and the LLM, just like the other works [1,2], as shown in Appendix B.1.
> > >
> > > **Information Leakage**
> > >
> > > As we stated before, we do not require the user to provide “explicit” feedback. We can extract the personalized triple from the conversations between the user and the LLM. Thus, there will be no more information involved in the optimization except the normal interactions between the user and the LLM, which is just like the other works [1,2] that are based on collecting user feedback.
> > >
> > > **Novelty**
> > >
> > > We want to emphasize that the contribution of our work is not to realize enhanced reasoning assisted by the KGs. The technical contribution and novelty of our work is to edit the KGs by constructing and adding personalized triples and remove conflicting and stale triples to realize efficient personalization based on the common interactions between LLM and the user.
> > >
> > > **Evaluation Metrics**
> > >
> > > Thanks for your insightful question. There are different names for the metric of “editing accuracy” in the baseline papers, and they are basically evaluating the same thing as the “efficacy” in the ROME paper. That is why we apply this metric for the “editing accuracy.” The reason why the baselines cannot achieve 100% editing accuracy is that we adopt the “long-term editing” setting, which is more practical for personalization. The evaluation setting of the compared knowledge editing baselines is to edit and evaluate only one sample each time, which is much easier. This is not the setting of our work. Our focus is to construct a personalized KG containing a large volume of personalized knowledge during long-term use, which means that multiple samples should be involved for each editing and evaluation. We also evaluate our method on the PeInt dataset, which is constructed based on real human-LLM conversations, which demonstrates the generalizability of our method.
> > >
> > > We understand that simply citing the paper [3] you mentioned in our current submission might not solve your concern about the contribution of our paper, so we explain our analysis by referring to the difference with theirs in detail and revise our paper to include this. Please see our reply to all.
> > >
> > > Please let us know if there is anything we can do to solve your following concerns.
> > >
> > > [1] Baek, J., Chandrasekaran, N., Cucerzan, S., Herring, A., & Jauhar, S. K. (2024, May). Knowledge-augmented large language models for personalized contextual query suggestion. In Proceedings of the ACM on Web Conference 2024 (pp. 3355-3366).
> > >
> > > [2] Madaan, Aman, et al. "Memory-assisted prompt editing to improve GPT-3 after deployment." arXiv preprint arXiv:2201.06009 (2022).
> > >
> > > [3] Luo, Linhao, et al. "Reasoning on Graphs: Faithful and Interpretable Large Language Model Reasoning." The Twelfth International Conference on Learning Representations.

---

> > > > ### Comment · Reviewer_kufp · 2024-11-27
> > > >
> > > > Thanks for the detailed response. However, I would keep my scores. Expect controversy about the theoretical analysis, the current version limits in the following aspects:
> > > >
> > > > 1) Limited novelty in methodology (retrieve facts from KGs for personalization/knowledge editing). Similar ideas have been proposed and not compared (see references in previous [comments](https://openreview.net/forum?id=oApCZZZ3O4&noteId=IKR7Ea3uWE)).
> > > > 2) Generability. The current experiment settings cannot well support the generability claims where the retrieved personalized knowledge can be long-term used for other tasks.

---

### Official Review · Reviewer_YoVu · 2024-11-03

**Soundness:** 1
**Presentation:** 2
**Contribution:** 1
**Rating:** 1
**Confidence:** 5

**Summary:**

This paper proposes a personalized knowledge editing algorithm for LLMs leveraging KGs. It avoids fine-tuning the LLMs while updating the user profile for new knowledge from feedback as a graph. The pipeline would be like: query->retrieve KG->LLM->user feedback->Update KG->LLM and iteratively. The idea is valuable for rapidly changing user preferences in various downstream tasks. However, the design of the methodology lacks careful consideration and the detailed comments can be found hereunder.

**Strengths:**

* This is indeed a Graph RAG paper and can be treated as a new angle which continuously updates the KG with new knowledge.
* The figures are well drawn and easy to follow.

**Weaknesses:**

* I recognize the merits of this paper by editing KGs for changing user preference. However, the problem of KG editing is not rocket science, especially each time there will only be very limited triples involved to be added or removed. It could be far more concisely and effectively solved with engineering techniques.
* The word 'tuning' is not suitable which may cause misunderstanding, especially when it only refers to changing the structure of the KG without any parameter learning.
* The code is very important to figure out how the structure of KGs is indeed edited, as well as for the probability calculation.
* **The paper heavily refers to ROG [1] for the entire methodology which involves plagiarism problem. It changes the storyline by introducing the concept of knowledge editing with no new methods/components.**  Therefore, there is no comment on the remaining of methodology.

[1] Reasoning on Graphs: Faithful and Interpretable Large Language Model Reasoning, ICLR' 24.

**Questions:**

N/A

**Details Of Ethics Concerns:**

* **The paper heavily refers to ROG [1] for the entire methodology which involves plagiarism problem. It changes the storyline by introducing the concept of knowledge editing with no new methods/components.**


[1] Reasoning on Graphs: Faithful and Interpretable Large Language Model Reasoning, ICLR' 24.

---

> ### Author Response · Authors · 2024-11-20
> **Rebuttal**
>
> Dear reviewer,
>
> I am replying to your complain about the code and charge of plagiarism.
>
> **Code:**
>
> We totally agree with you about the importance of submitting the code. We had already submitted the code to the supplementary material before the submission deadline.
>
> **Plagiarism:**
>
> We appreciate the time and effort you dedicated to reviewing our work. However, we want to remind you that allegations of plagiarism are serious and should be supported by clear evidence. While constructive reminders of related work are always welcome, unfounded accusations undermine the professionalism and respect integral to the peer-review process.
> We respect your anonymity but firmly reject your claim of plagiarism. If you believe we have violated academic integrity, we suggest you raise this matter with the Area Chair (AC) or Program Chair (PC), even though I believe they have noticed your concern, for an impartial evaluation.
>
> Sincerely,
>
> Authors of submission 8863

---

> > ### Comment · Reviewer_YoVu · 2024-11-20
> >
> > Thanks for the authors' responses.
> >
> > I would like to say you are lucky since it seems like other reviewers happen not to have read the ROG paper which you have heavily referred to (I will avoid using the p-word and let others define your paper since the comments will be publicly available).
> >
> > * You request the proof while I have already cited ROG in my comments, everyone who takes a look at the paper will definitely get the point.
> > * You claim big and strongly for the p-word problem, but **no** any words to rebuttal about the ROG paper. Would you mind if you could explain more about the differences between yours and ROG? since you 'learned' a lot from it and **no** even one reference in the paper?
> > * Except for the contributions that are made in **ROG** paper, I have acknowledged your good points of KG editing. However, as commented before, it is no rocket science at all, nor enough to support your paper to be a high-quality top-conference paper. And you have no rebuttal about the first two comments.
> >
> > Therefore, I would strongly **reject** this paper.

---

### Official Review · Reviewer_KoUf · 2024-11-03

**Soundness:** 2
**Presentation:** 3
**Contribution:** 2
**Rating:** 5
**Confidence:** 2

**Summary:**

This paper presents Knowledge Graph Tuning (KGT), a novel method for personalizing large language models (LLMs) by leveraging knowledge graphs (KGs) rather than modifying model parameters. KGT captures personalized factual knowledge from user interactions, updates knowledge graphs, and optimizes them without requiring back-propagation, thereby enhancing computational and memory efficiency.

**Strengths:**

I. Efficiency: KGT reduces latency and GPU memory usage by eliminating the need for back-propagation, making it suitable for real-time applications.

II. Interpretability: By using KGs to manage user knowledge, KGT enables transparent adjustments to user-specific information, enhancing the interpretability of the personalized responses.

**Weaknesses:**

I. KGT shares certain similarities with MemPrompt (https://arxiv.org/pdf/2201.06009). The authors should provide a detailed comparison to highlight the distinctive aspects of their approach.

II. Dependence on KG Quality: The effectiveness of the method relies on the accuracy and relevance of the KG. Low-quality KGs may lead to less effective personalization.

III. Complexity of Knowledge Retrieval: KGT may struggle with more complex or abstract knowledge retrieval tasks, where simple knowledge triples may not adequately represent user-specific knowledge.

IV. The quality of the presentation is below ICLR 2025 standards. For example:

 a. The format of the references should be consistent to ensure neatness and professionalism. For instance, the names of conferences should be uniformly presented either in abbreviations or full names, rather than a mixture of both.

b. Some of the referenced papers have already been accepted but are still listed as preprints on arXiv. For example, the line 661 should be "Zhengyan Zhang, Xu Han, Zhiyuan Liu, Xin Jiang, Maosong Sun, and Qun Liu. 2019. ERNIE: Enhanced Language Representation with Informative Entities. In Proceedings of the 57th Annual Meeting of the Association for Computational Linguistics, pages 1441–1451, Florence, Italy. Association for Computational Linguistics."

**Questions:**

Please see weakness

---

> ### Author Response · Authors · 2024-11-20
> **Rebuttal**
>
> We thank the reviewer for acknowledging that our method is efficient and interpretable. Upon the concerns of the reviewer, we will address them one by one.
>
> **W1: detailed comparison with MemPrompt.**
>
> We thank the reviewer for drawing attention to MemPrompt. Our work possesses distinct differences, detailed as follows:
> 1. Storage: Memprompt uses a dictionary where keys and values are input and user feedbacks, respectively. Both of them are strings in natural language. We extract and save the knowledge triples, enjoying advantages that a. We use less memory. b. We enjoy certain level of abstraction that allows better generalizability.
> 2. Dynamic updating on conflicting knowledge: Memprompt has trouble resolving conflicts (also mentioned in the limitation of the paper). It is the focus of our paper to detect conflicting knowledge facts during the LLM-AI interaction and update the stale knowledge in the knowledge base.
> 3. Retrieval augmentation approach: Memprompt directly concatenate the user feedback to the input to the LLM once the input triggers the additional memory. However, we employ a knowledge graph as additional enhancement for reasoning, providing a more structured augmentation.
> 4. Theoretical motivation: Memprompt largely relies on the in context understanding ability of the GPT3 model. Meanwhile, we provide a theoretical insights to motivate our approach under the framework of RAG.
> We have revised our draft to include this reference and discussion in related works.
>
>
> **W2: reliance on the quality and completeness of the knowledge graphs.**
>
> Instead of heavily relying on the quality of KGs, our algorithm is more of a method to improve the quality and completeness of the knowledge graph during the daily use of LLMs. Even if the KG is low-quality and incomplete, our method can inject and update the knowledge triples in the graph automatically based on the user’s interaction with the LLM.
> In addition, with the development of LLM, people realize that the structural knowledge base, such as knowledge graphs, is valuable to enhance LLM's reasoning. More and more knowledge graphs are developed and proven effective in cooperating with LLMs [1-3], including commonsense knowledge graphs[4] and some domain-specific graphs like Medical Knowledge Graphs (MKG) [5,6]. We believe that the quality and quantity of KG will improve continually, especially with the rapid development of LLMs and knowledge-enhanced inference technologies. [References sees reply for reviewer 5.]
>
> **W3: KG on more complex tasks.**
>
> We only focused on the one-hop setting (knowledge triples) in this paper because this is the first paper to adapt the knowledge graph to realize personalization. A multi-hop solution (more relations to detangle more complex tasks) is definitely a goal we aim to achieve for future research. However, we want to mention that even though the detailed techniques might be different, the general theoretical framework supports multi-hop solutions. We are delighted to provide the idea to realize the multi-hop KG editing.
>
> When we need to edit the KG in the multi-hop setting, we can still utilize Equation 2 since it does not have restrictions on the one-hop setting. The difference is the sampling of $z$, $Q(z|q,a)$, and $\mathcal{H}(q,a,K)$. There should be a constant $I$ to define the most hops to cover in the editing. Then $z$ can be formulated as $(e_q, r_k^1, e_k^2,..., r_k^i, e_a )$ where $i\leq I$. When we need to cover the range of $I$ hops in the KG, we will sample $K$ $z$s for each $i\leq I$. We will use $\mathcal{H}(q,a,K,I)$ to denote the set of personalized triples, where $$ \mathcal{H}(q,a,K,I) = \bigcup_{i=1}^I\mathcal{H}^i(q,a,K), $$ and $\mathcal{H}^i(q,a,K)={(e_q, r_k^1, e_k^2,..., r_k^i, e_a )}_{k\in [K]}$ is the set of personalized paths with $i$ hops. Then, $Q(z|q,a)$ is approximated as
>
> $$ Q(z|q,a) \simeq \frac{1}{KI} \text{ if } z \in \mathcal{H}(q,a,K,I) \text{ else } 0. $$
>
> Then, we replace the $\mathcal{H}$ and $K$ in Equation 7 to derive the final loss function
>
> $$ \mathcal{L} = \mathcal{L_{\text{retrieve}}} + \mathcal{L_{\text{reasoning}}} = -\frac{1}{KI}\sum_{z\in \mathcal{H}(q,a,K,I)} \log \left[P_{\theta,\mathcal{G}}\left(a | q,z\right)P_{\theta,\mathcal{G}}\left(z | q\right)\right]. $$
>
> The optimization method can still be applicable to the multi-hop setting. The detailed prompts to derive personalized $z$ and the computation of $P_{\theta,\mathcal{G}}\left(z|q\right)$ need to be adjusted, but the basic idea is the same with the one-hop setting.
>
> Our primary focus in this paper is to prove that editing KG to realize LLM personalization works. However, we realize that a multi-hop solution will definitely advance our algorithm under more complex settings, and we will include the discussion about the detailed multi-hop idea in future work!
>
> **W4: reference format.**
>
> We thank the reviewer for pointing this out. We have revised the reference list to make them consistent and updated.

---

> > ### Comment · Reviewer_KoUf · 2024-12-03
> > **Response to the authors**
> >
> > Thank you for your response! I think I will keep my rating.

---

### Official Review · Reviewer_jsgs · 2024-11-04

**Soundness:** 2
**Presentation:** 3
**Contribution:** 3
**Rating:** 6
**Confidence:** 5

**Summary:**

The authors propose to personalize the outputs of LLMs by augmenting them with the triplets from the knowledge graph (that is constructed based on the interactions between humans and LLMs), without fine-tuning. Specifically, the authors iteratively update the knowledge graph by generating the triplet that leads to the high probability of generating the correct personalized answer while removing the one with the lowest probability, and then use this constructed knowledge graph by retrieving the relevant triplets from it in response to the given query and augmenting LLMs with them in generating the answer. The authors validate the proposed approach on two knowledge editing benchmark datasets as well as real-world interaction data between humans and LLMs collected from five people, on which it outperforms relevant baselines.

**Strengths:**

* The proposed approach of updating the triplets within the knowledge graph in regards to the probability of generating the correct answer with them is sound and reasonable.
* The proposed approach outperforms relevant baselines substantially, while only requiring low computational costs.

**Weaknesses:**

* The authors do not provide details about how to collect the interaction data between humans and LLMs, for example, how many samples are annotated (i.e., the number of samples seems not comprehensive enough), who are the annotators, and how to pay them. In addition to this, the authors may further demonstrate the quality of the annotated data, as currently it seems questionable.
* Besides this human-annotated data, the authors validate the proposed approach on knowledge editing benchmark datasets (whose goal is to generate the outputs that are counter intuitive compared to the knowledge that common LLMs or knowledge graphs have), which may not be appropriate to evaluate the performance of LLM personalization approaches.

**Questions:**

Please see the weaknesses above.

---

> ### Author Response · Authors · 2024-11-20
> **Rebuttal**
>
> We sincerely appreciate the reviewer's positive evaluation of our work. Specifically, the reviewer recognizes the soundness and reasonableness of our approach. Moreover, the reviewer acknowledges that our method surpasses the established baselines while incurring lower computational costs.
>
> Regarding the reviewer's concern:
>
> **W1: more details of the interaction dataset.**
>
> We engaged five real users through our university's volunteer recruitment portal. These volunteers interacted with the LLM (Llama3-7b) on various topics tailored to their personal preferences and experiences. One example of interaction is like
>
> `User 1: "Can you update me on the latest publications related to quantum entanglement from the past month and also help me plan my day tomorrow? I have a full day at the lab."`
>
> `AI Assistant: "I found five new publications that match your criteria, including two from the Quantum Information Journal. Would you like a summary of each, or should I focus on those with experimental results similar to your project? Also, I see you have a busy day at the lab tomorrow. There's flexibility in your afternoon schedule; how about scheduling some violin practice for the evening?"`
>
> `User 1: "Please summarize the ones with experimental results, and download the full papers for later. Schedule violin practice at 7:00 PM and remind me to take Schrödinger for his vet appointment."`
>
> `AI Assistant: "I've scheduled violin practice for 7:00 PM and set a reminder for Schrödinger’s vet appointment at 3:00 PM, with a 30-minute buffer for travel. The summaries of the selected publications will be ready in your inbox within the hour. Is there anything else I can assist with?"`
>
> `User 1: "Thanks! Could you also check for any chess tournaments happening this weekend? I'd like to participate if possible."`
>
> `AI Assistant: "I will look into the local chess club listings and let you know about the tournaments this weekend. Anything else you need?"`
>
> There are several personal factual knowledge contained in this example, including that user 1's research field is related to quantum physics, user 1's pet is called Schrödinger, and user 1's hobby is chess. We collect all these interactions and evaluate if our method and the baselines can adapt to these personalized knowledge bases.
>
> **W2: concerns on using knowledge editing benchmark dataset to evaluate personalization.**
>
> Our algorithm is to realize personalization by editing factual knowledge in the KG. So, generally, our algorithm is also a method of knowledge editing technique. Counterfactual knowledge is the "extreme and difficult case'' of personalized knowledge, and that is why we apply the counterfactual knowledge editing benchmark to evaluate our method. Further, we thoroughly considered concerns as such. Consequently, we built a real personalization interaction benchmark.

---

> ### Comment · Reviewer_jsgs · 2024-11-30
>
> Thank you for your response. I have read it carefully and I still believe that the main evaluation datasets (about editing factual knowledge in KGs) are not perfectly aligned with the goal of LLM output personalization. Therefore, I will keep my score.
>
> P.S. I have read the reviews from other reviewers including Reviewer YoVu, and (while I see some clear differences between the ROG paper and this work) I hope the authors will more carefully discuss this previous work in the revision.

---

### Author Response · Authors · 2024-11-20
**Distinguishing our analysis and contributions**

Dear all,

While we do not accept the accusation of plagiarism by one of the reviewers and will let PCs and ACs solve this issue, we sincerely appreciate all your effort in reviewing our paper and are respectfully willing to solve the concern in distinguishing our contribution and theoretical analysis from the related work RoG[1]. We understand that simply citing their paper in our current submission might not solve this concern, so we explain our analysis by referring to the difference with theirs in detail here, even though we did not develop our analysis based on their paper. Before the detailed comparison, we want to emphasize that the goal of their paper is to train the LLM to better reason on the KG, while our goal is to edit the KG to realize more efficient and interpretable personalization, which is totally different.

We first derive the optimization objective (Equation 2 in our submission) by following the common practice of RAG-based analysis [2,3]. They have a similar equation in their paper, which is normal since this derivation is commonly applied in the community. However, it is notable that we achieve the optimization objective by optimizing $\mathcal{G}$, while they achieve the objective by optimizing $\theta$, as indicated in the difference of Equations in two drafts. In addition, their latent parameter $z$ is a plan, while ours is a knowledge triple. There are three terms to formulate to calculate all the ELBO-derived retrieval-based objectives: posterior distribution, retrieval probability, and reasoning probability, which are all different in our submission and their paper.

When formulating the posterior distribution, we estimate the distribution on different sampling spaces. They conduct optimization on a fixed knowledge graph, which allows them to calculate the distribution by traversing the existing graph. We conduct optimization on the KG, and we calculate the distribution by considering all the potential knowledge triples, even if they are not involved and intractable in the current KG. Thus, we design a method to formulate the dynamic sampling space, which includes the potential personalized knowledge triples. The calculations of the retrieval probability are different because the sampling spaces and the latent parameters $z$ are totally different in the two papers, which is also the case when calculating the reasoning probability.

To optimize the objective, we edit the knowledge graph while they train the model parameters, which are totally different.

You are also welcome to check our code, and you will find that nothing in our code has been developed based on their code.

**Thus, the goals, objective formulation, optimization methods, and code of the two papers are totally different.** We have revised the paper to highlight the difference between the two papers out of respect for the reviewers and to solve any of your concerns.

Sincerely,

Authors of submission 8863

[1] Luo, Linhao, et al. "Reasoning on Graphs: Faithful and Interpretable Large Language Model Reasoning." The Twelfth International Conference on Learning Representations.

[2] Lewis, Patrick, et al. "Retrieval-augmented generation for knowledge-intensive nlp tasks." Advances in Neural Information Processing Systems 33 (2020): 9459-9474.

[3] Paranjape, Ashwin, et al. "Hindsight: Posterior-guided training of retrievers for improved open-ended generation." International Conference on Learning Representations (2022).

---

> ### Comment · Area_Chair_9m69 · 2024-11-25
> **Please reply to the authors' response.**
>
> Dear reviewers,
>
> The ICLR author discussion phase is ending soon. Could you please review the authors' responses and take the necessary actions? Feel free to ask additional questions during the discussion. If the authors address your concerns, kindly acknowledge their response and update your assessment as appropriate.
>
>
> Best,
> AC

---

### Meta-Review · Area_Chair_9m69 · 2024-12-13

**Metareview:**

The paper proposes a method to enhance user experience by personalizing large language models (LLMs) in real-time using Knowledge Graphs (KGs). The authors introduce Knowledge Graph Tuning (KGT), which extracts personalized knowledge from user interactions and optimizes KGs without modifying LLM parameters, aiming to improve computational and memory efficiency, ensure interpretability, and reduce latency.

There is discussion highlighting the paper’s strengths, including its efficiency, interpretability, scalability, and sound experimental validation. However, concerns were raised about the dependence on KG quality, limited novelty, and handling contradictory user feedback. Ethical concerns were also mentioned due to similarities with a previous paper (RoG). The authors defended their work by clarifying differences from the RoG paper, emphasizing unique contributions, and addressing concerns about KG quality and user feedback.

The AC checked both papers and did see the differences between them. However, a careful discussion of this previous work would be beneficial in a revised version. I suggest the authors take this discussion and other concerns (including novelty and generalization) into consideration to improve their paper and resubmit it to another venue.

**Additional Comments On Reviewer Discussion:**

There is discussion highlighting the paper’s strengths, including its efficiency, interpretability, scalability, and sound experimental validation. However, concerns were raised about the dependence on KG quality, limited novelty, and handling contradictory user feedback. Ethical concerns were also mentioned due to similarities with a previous paper (RoG). The authors defended their work by clarifying differences from the RoG paper, emphasizing unique contributions, and addressing concerns about KG quality and user feedback.

---

### Decision · Program_Chairs · 2025-01-22

Reject